# Keto-anthraquinone covalent organic framework for $H_2O_2$ photosynthesis with oxygen and alkaline water

Xiangcheng Zhang[1], Silian Cheng[1], Chao Chen [2], Xue Wen[1], Jie Miao[1], Baoxue Zhou [1], Mingce Long [1]✉ & Lizhi Zhang [1]✉

Hydrogen peroxide photosynthesis suffers from insufficient catalytic activity due to the high energy barrier of hydrogen extraction from $H_2O$. Herein, we report that mechanochemically synthesized keto-form anthraquinone covalent organic framework which is able to directly synthesize $H_2O_2$ (4784 μmol $h^{-1}$ $g^{-1}$ at λ > 400 nm) from oxygen and alkaline water (pH = 13) in the absence of any sacrificial reagents. The strong alkalinity resulted in the formation of $OH^-(H_2O)_n$ clusters in water, which were adsorbed on keto moieties within the framework and then dissociated into $O_2$ and active hydrogen, because the energy barrier of hydrogen extraction was largely lowered. The produced hydrogen reacted with anthraquinone to generate anthrahydroquinone, which was subsequently oxidized by $O_2$ to produce $H_2O_2$. This study ultimately sheds light on the importance of hydrogen extraction from $H_2O$ for $H_2O_2$ photosynthesis and demonstrates that $H_2O_2$ synthesis is achievable under alkaline conditions.

Hydrogen peroxide ($H_2O_2$), a chemical with increasing market share, finds extensive applications in biomedicine, disinfection, bleaching, organic synthesis, and water treatment[1-4]. The well-known industrial production of $H_2O_2$ is the anthraquinone (AQ) process, which suffers from intensive energy consumption and waste discharge[5]. As a green and carbon-neutral alternative, solar driven oxygen reduction strategy of $H_2O_2$ synthesis from molecular oxygen and water attracts more and more attention[6-9]. Although many photocatalysts are effective for the $H_2O_2$ synthesis, high dosages of organic sacrificial reagents such as isopropanol are always used to scavenge photogenerated holes and offer hydrogen for the $H_2O_2$ formation, which obviously bring in undesired impurity and also increase the cost of $H_2O_2$ synthesis[10]. In comparison with organic sacrificial reagents, water is more inexpensive and convenient hydrogen source, but an intrinsically poor hydrogen donor, because water molecules have a high O-H bond dissociation energy (BDE, 492 kJ $mol^{-1}$ for homolytic cleavage)[11,12]. Thus, highly efficient $H_2O_2$ photosynthesis only with molecular oxygen and water is of great significance, but remains a giant challenge.

It is well known that the hydrogen-bond (H-bond) in adsorbed water clusters plays critical role on water dissociation during photocatalysis[13,14]. At a "pseudodissociated" state[14], the intermolecular H-bond facilitates the cleavage of water O-H bond at <1 monolayer coverage[15], and interface H-bond also promotes photogenerated hole transfer and water oxidation by the strongly coupling of H-bond with holes[16]. Unfortunately, strong hydrogen bond network among water clusters inhibits the water dissociation[17]. Thus, an accurate control of H-bond network and adsorbed water monolayers over photocatalysts is vital for water dissociation[13]. Recently, scientists found that excess electrons of $OH^-$ anions in alkaline water could induce reorganization of hydrogen bond in water clusters, thus further diminishing the overall energy barrier of alkaline hydrogen evolution reaction (HER)[18]. However, it is still unknown whether this alkaline based H-bond network manipulation strategy is feasible for the $H_2O_2$ photosynthesis.

Different from traditional metal oxide photocatalysts of poor interfacial H-bond modulation capacity, covalent organic frameworks (COFs), famous metal-free molecular photocatalysts possessing huge

[1]School of Environmental Science and Engineering, Shanghai Jiao Tong University, Shanghai 200240, China. [2]School of Ecological and Environmental Science, Key Laboratory for Urban Ecological Processes and Eco-Restoration, East China Normal University, Shanghai 200241, China. ✉e-mail: long_mc@sjtu.edu.cn; zhanglizhi@sjtu.edu.cn

potential in $H_2O_2$ photosynthesis, are very powerful to regulate H-bond at molecular levels because of their variable and designable organic units[19,20]. Among various organic units, AQ moieties is believed to be the optimal redox center for the $H_2O_2$ synthesis, as the oxidation of the hydrogenated AQ (anthrahydroquinone, HAQ) by molecular oxygen can selectively produce $H_2O_2$, which is thermodynamically spontaneous and commercially used[5]. Recently, several AQ-containing COFs (such as TPE-AQ, TpAQ, AQTEE-COP, and AQTT-COP) were designed to promote photogenerated charge separation and facilitate WOR for efficient $H_2O_2$ photosynthesis with pure water upon visible light irradiation (>400 nm), and their best activity reached 3221 μmol $g^{-1}$ $h^{-1}$ without manipulating H-bond network[21–24].

As a typical AQ-containing COFs, TpAQ synthesized by β-ketoenamines links of 2,6-diaminoanthraquinone (AQ) and 2, 4, 6-triformylphloroglucinol (Tp), is often a mixture of keto- and enol-forms due to the formation of tautomerism during the polymerization (Fig. S1). Different from the unstable enol-form that mainly form weak H-bond with oxygen in $H_2O$[25], keto-form AQ COFs (Kf-AQ) is a more favorable proton acceptor to combine with hydrogen in $H_2O$ via strong H-bond. Generally, traditional solvothermal method with acetic acid catalysis tends to produce enol-form dominant COFs. Although alkaline solution (such as OH⁻) induces the transformation of enol-form into keto-form[26], the NaOH addition disfavored the solvothermal synthesis of Kf-AQ, because the excessive solvents would consume NaOH to form carboxylates. Thus, the controlled synthesis of Kf-AQ is crucial for $H_2O_2$ photosynthesis, but never reported previously.

Herein we demonstrate the mechanochemical synthesis of keto-form anthraquinone covalent organic framework (Kf-AQ) for direct $H_2O_2$ photosynthesis with molecular oxygen and alkaline water (pH = 13), and this Kf-AQ could deliver a record $H_2O_2$ production rate of 4784 μmol $h^{-1}$ $g^{-1}$ in the absence of any sacrificial reagents under visible light irradiation (λ > 400 nm). The critical roles of hydroxide anions and keto-form AQ moieties for efficient $H_2O_2$ production are carefully clarified via in-situ characterization and theory calculations.

## Results and discussion

### Synthesis and structure characterization

Kf-AQ was mechanochemically synthesized by a Schiff-base condensation reaction of Tp and AQ with $CH_3COONa$ (NaAc) as the catalysts (Fig. 1a). Fourier-transformed infrared spectra (FT-IR) spectra clearly revealed a new C-N stretching band at 1260 $cm^{-1}$ and a disappeared N-H stretching band at 3459-3151 $cm^{-1}$ for $NH_2$ groups in AQ (Fig. S2)[27]. The as-prepared Kf-AQ powder displays a red-black color, corresponding to its wide optical absorption with the edge extended to 900 nm (Fig. S3), which is obviously red-shifted as compared to the absorption edge at 780 nm of TpAQ prepared by a traditional solvothermal method[28]. The simulated powder X-ray diffraction (PXRD) pattern of Kf-AQ with eclipsed AA stacking mode, whose fractional atomic coordinate data for the unit cell were presented in Table S1, agreed with the experimental data in a large extent (Fig. 1b), suggesting the validity of such structure in Kf-AQ. Particularly, the broad peak at 26.54° was caused by the strong π-π stacking construction arisen from the existence of a multilayered COF structure with an interlayer distance of 3.48 Å. TEM and SEM images also displayed that Kf-AQ had a lamellar stacking structure and excellent crystallinity with an observable 0.33 nm lattice spacing (Fig. 1c, d and Fig. S4)[28],very close to the simulated interlayer spacing (0.348 nm). Moreover, Kf-AQ had a

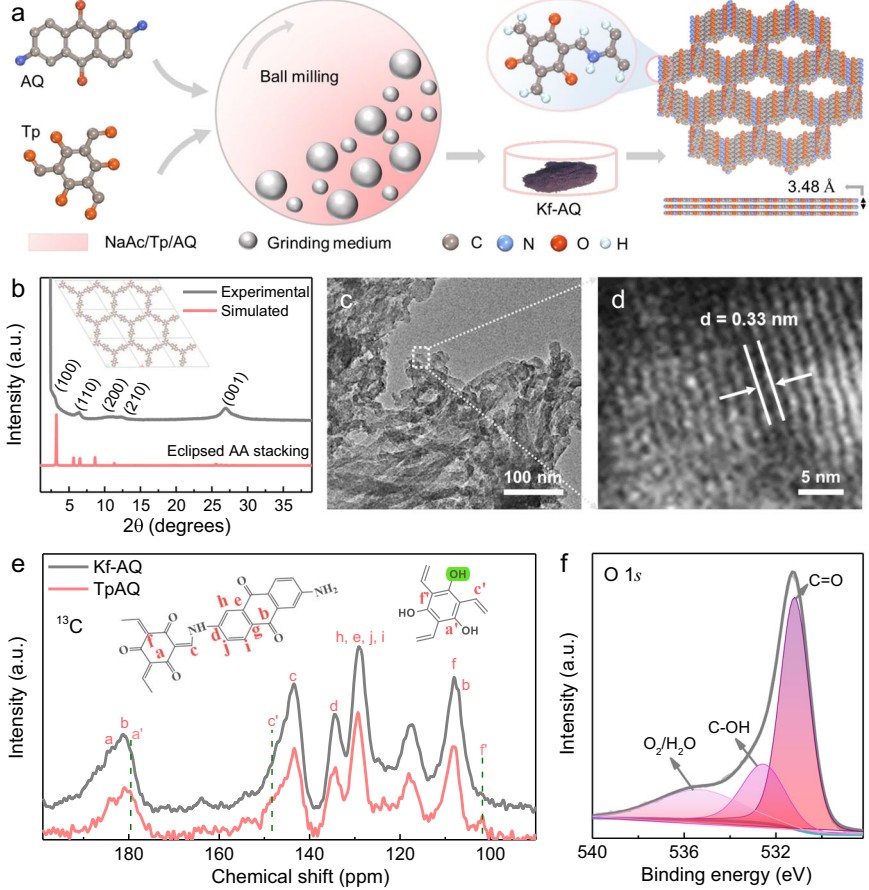

**Fig. 1 | Synthesis process and structural characterization. a** Schematic of Kf-AQ condensation. **b** PXRD patterns of Kf-AQ, experimentally observed (dark) and simulated using eclipsed AA-stacking (red), the inset is the crystal structures of the eclipsed AA stacking model, the simulated cell parameters of (**a**, **b** and **c**) are 30.59, 30.59 and 3.51 Å, respectively. **c**, **d** TEM images of Kf-AQ. **e** $^{13}C$ CP-MAS solid state NMR spectra of Kf-AQ and TpAQ. **f** High-resolution O1s XPS spectra for Kf-AQ.

specific surface area of 141.7 m$^2$ g$^{-1}$ and a pore size of 2.2 nm (Fig. S5), which was well matched with the simulated value (2.28 nm) in Fig. 1a.

Solid-state NMR spectra revealed that Kf-AQ had an almost exclusive keto-form structure (Fig. 1e). The chemical shifts at 184 ppm, 144 ppm and 109 ppm in $^{13}$C NMR spectra were all indexed to the keto-form structure of Kf-AQ[25,27,29], while the chemical shift at 179 ppm (a') and 101 ppm (f') for the enol-form carbon and at 4.4 ppm for the enol-form hydrogen (c, C-OH) was absent in the $^{13}$C and $^1$H NMR spectra, respectively (Fig. 1e and Fig. S6)[30]. Only C, O and N elements were present in the XPS survey spectrum of Kf-AQ, without any residual Na (Fig. S7). More evidences of keto-form structure in Kf-AQ could be found in high resolution XPS spectra (Fig. S8). The content of C=O in Kf-AQ was approximately 60%, about three times that of C-OH (22%) (Fig. 1f). The content of C-N-H (56%), corresponding to the deconvolution peak at the binding energy of 400.4 eV in N 1$s$ spectra, was obviously higher than that of C=N (31%) (Fig. S9a)[31], while C 1$s$ XPS spectra also illustrated more C-C (52%) and less C-O (38%) in Kf-AQ (Fig. S9b)[32]. All these above results supported the successful synthesis of an exclusive keto-form AQ COF.

The formation of Kf-AQ might be attributed to a NaAc-catalyzed Schiff-base condensation process as follows. Upon the heat generated from the collision of balls, the carbonyl oxygen on Tp monomer undergoes a nucleophilic addition with Na$^+$ to form aldehyde salts[33,34], resulting in the neighbor carbon acquiring a positive charge to fulfill another nucleophilic addition with nitrogen atoms in AQ. The generated α-hydroxyl undergoes further dehydration with adjacent amino hydrogen to form an enol-form COF. Subsequently, Ac$^-$ anions as the Lewis base tend to bind with the hydrogen of hydroxyl group in enol moieties and then induce electron transfer from oxygen to alkene group, thus enabling the enol-form transformation into the thermodynamically more stable keto-form moiety (Fig. S10). Such a transformation cannot be driven in the traditional solvothermal synthesis, but might partially occur in alkaline water to produce a keto-form dominated AQ COF[25,35].

## Efficient H$_2$O$_2$ photosynthesis

The H$_2$O$_2$ photosynthesis performance of Kf-AQ was evaluated by dispersing the powder in water at neutral and alkaline solutions (pH = 9, 11, 13, 14) with continuous O$_2$ purging. Upon visible light (λ > 400 nm) irradiation, the rate of H$_2$O$_2$ production at pH 13 reached as high as 4784 μmol h$^{-1}$ g$^{-1}$ (Fig. 2a), a record in H$_2$O$_2$ photosynthesis of AQ containing COFs with water (Fig. 2b)[21-23,36-56]. Upon a prolonged irradiation for 5 h, the H$_2$O$_2$ production was steadily growing (Fig. S11), and kept constant during five cycles of reaction (Fig. 2c). The crystal structure and surface functional groups of the reacted Kf-AQ did not change (Fig. S12), demonstrating its excellent stability for the H$_2$O$_2$ photosynthesis. The contribution of Na$^+$ to the enhanced H$_2$O$_2$ production was ruled out by the replacement of NaOH with NaCl and KOH (Fig. S13), confirming the crucial promoting effect of hydroxide anions on the H$_2$O$_2$ photosynthesis of Kf-AQ.

The kinetic of H$_2$O$_2$ production was analyzed by fitting the time-dependent H$_2$O$_2$ production curves (Text S2). Kf-AQ exhibited the highest H$_2$O$_2$ formation rate constant ($k_f$, 31.39 μM min$^{-1}$), but a medium decomposition rate constant ($k_d$) (0.031 μM min$^{-1}$) at pH 13 (Fig. 2d and Table S2). Thus, the high H$_2$O$_2$ photosynthesis performance of Kf-AQ with alkaline water was mainly attributed to its better H$_2$O$_2$ formation ability. The apparent quantum efficiencies (AQY) of Kf-AQ at different wavelengths were well matched with its absorption spectrum, and the highest value appeared at 400 nm and reached 15.8% (Fig. 2e and Table S3). To the best of our knowledge, the AQY of Kf-AQ is higher than those of most reported H$_2$O$_2$ synthesis photocatalysts[57,58]. The solar-to-chemical conversion (SCC) efficiency of Kf-AQ was estimated to be 0.70% at pH 13 (Fig. S14 and Table S4), which was almost seven times of the average solar-to-biomass conversion (SBC) efficiency in nature[23].

## Mechanism investigation

We first checked the basic semiconductor properties of Kf-AQ to understand its high performance in H$_2$O$_2$ photosynthesis. The Tauc plot showed that the band gap of Kf-AQ was 1.55 eV (Fig. S15a), and ultraviolet photoelectron spectroscopy (UPS) determined its valence band potential ($E_{VB}$) as 1.90 V (Fig. S15b), suggesting that the conduction band potential ($E_{CB}$) of Kf-AQ was accordingly calculated as 0.35 V. Therefore, both 2e$^-$ ORR (0.68 V vs. RHE) to produce H$_2$O$_2$ and 4e$^-$ WOR (1.23 V vs. RHE) to evolute O$_2$ were thermodynamically feasible for Kf-AQ photocatalysis (Fig. 3a)[59]. We further conducted density functional theory (DFT) calculations to elucidate the exciton dissociation in photocatalysis by using the dimer models of Kf-AQ and enol-form TpAQ. As depicted in Fig. 3b, the highest occupied molecular orbital (HOMO) of Kf-AQ dimer uniformly disperses in the whole structure, while the lowest unoccupied molecular orbital (LUMO) mainly localized at AQ units, without any residual LUMO on the benzene ring of the Tp moiety. Thus, the HOMO-LUMO transition under excitation can redistribute electron density from the Tp moieties to the adjacent AQ units, thus resulting in effective intramolecular charge transfer in Kf-AQ. In the contrast, the HOMO of enol-form TpAQ is uniformly distributed on the dimer, while the LUMO mainly localizes at the AQ moiety and overlaps with the HOMO, with a portion of LUMO remaining on the benzene ring of the Tp moiety (Fig. S16). These results indicate that Kf-AQ is more favorable for the separation and transfer of photogenerated charges. The fluorescence spectra of TpAQ and Kf-AQ provided further evidence for their charge separation performance. Kf-AQ displayed much weaker fluorescence intensity than TpAQ in the steady-state fluorescence spectra (Fig. S17a), and the time-resolved fluorescence analysis showed that Kf-AQ had a longer relaxation time of electrons (6.82 ns) than TpAQ (6.07 ns) (Fig. S17b), thus proving the better performance of Kf-AQ in photogenerated-charge separation.

We then explored the sources of H and O for the H$_2$O$_2$ production by various control experiments and isotopic labeling analysis. In comparison to oxygen atmosphere, either air or N$_2$ purging resulted in poor H$_2$O$_2$ production (Fig. 3c), and the O$_2$ concentration in an airtight oxygen saturated suspension decreased obviously during photocatalysis (Fig. S18), suggesting the dominated contribution of ORR to the H$_2$O$_2$ production. AgNO$_3$ was added as the electron scavenger in N$_2$ atmosphere to evaluate the contribution of water oxidation. The negligible amount of H$_2$O$_2$ generated in the Kf-AQ suspension ruled out the direct contribution of WOR to the H$_2$O$_2$ production. However, H$_2$O$_2$ was obviously produced in case of N$_2$ purging and absence of AgNO$_3$, suggesting that photocatalytically produced O$_2$ via 4e$^-$ WOR (Fig. 3c) enabled the consequent ORR to produce H$_2$O$_2$ (Fig. S19). Significant H$_2$O$_2$ was only detected in the mixed solution of H$_2$O and acetonitrile ($v/v$ = 1:1) other than pure acetonitrile (Fig. 3d), confirming that water was the exclusive hydrogen source for the H$_2$O$_2$ photosynthesis.

We conducted the isotopic photoreaction experiments by purging H$_2$$^{16}$O suspensions with $^{18}$O$_2$ gas during the H$_2$O$_2$ photosynthesis, and then used MnO$_2$ to catalytically decompose the as-synthesized H$_2$O$_2$ into oxygen. After 8 h of photoreaction, strong $^{18}$O$_2$ ($m/z$ = 36, 93.7%) and very weak $^{16}$O$_2$ ($m/z$ = 32, 6.3%) signals appeared in the gas chromatography-mass spectra (GC-MS) of collected gas (Fig. 3e), demonstrating that H$_2$$^{18}$O$_2$ was the dominated product and mainly came from the reduction of $^{18}$O$_2$. Gradually, the signal of $^{18}$O$_2$ peak decreased (80.5%), accompanying with an increased $^{16}$O$_2$ signal (19.5%) at 24 h of reaction, because the photocatalytic oxidation of H$_2$$^{16}$O produced $^{16}$O$_2$ to increase the proportion of H$_2$$^{16}$O$_2$ in the products. The electrons transfer number ($n$) of ORR was further measured to be about 2.06 - 2.09 by the RDE method (Figs. 3f and S20)[24]. Thus, we suppose that both 2e$^-$ ORR and 4e$^-$ WOR take place during H$_2$O$_2$ photosynthesis over Kf-AQ at pH 13.

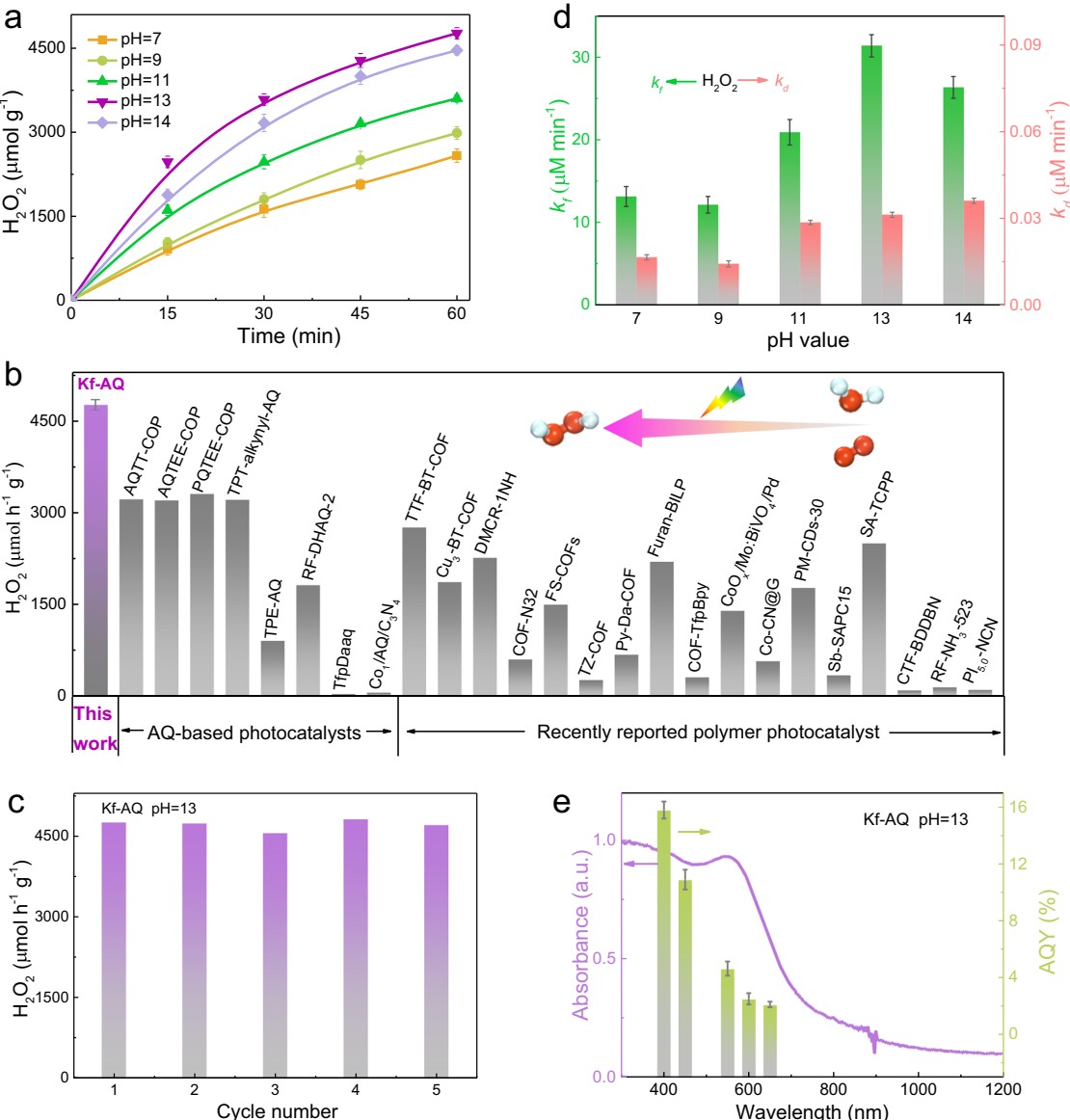

**Fig. 2 | $H_2O_2$ photosynthesis. a** Photocatalytic $H_2O_2$ production at different pH conditions. Error bars are the standard deviations of three replicate measurements. **b** A comparison of photocatalytic $H_2O_2$ production rates for photocatalysts in the absence of sacrificial regents. **c** The recycling tests of Kf-AQ at pH = 13. **d** The rate constants of $H_2O_2$ formation ($k_f$) and decomposition ($k_d$). **e** Wavelength-dependent AQY of photocatalytic $H_2O_2$ production on Kf-AQ at pH = 13.

To probe the active sites of Kf-AQ for the $H_2O_2$ photosynthesis, we synthesized two control COFs by respectively replacing the monomers of Tp and AQ with 1,3,5-trimethylbenzaldehyde (LZU) and 2,6-diaminoanthracene (DA), namely LZUAQ and TpDA (Figs. S21, S22). Their $H_2O_2$ photosynthesis performance was much worse than that of Kf-AQ (Fig. S23), suggesting that anthraquinone groups were the indispensable active sites for ORR, and the keto and AQ conjugated configuration accounted for the efficient WOR over Kf-AQ.

We then employed in-situ FTIR and Raman spectra to further understand the critical role of water adsorption and dehydrogenation in the $H_2O_2$ photosynthesis of Kf-AQ. Upon irradiation, three obvious O-H stretching vibration bands appeared in the in-situ FTIR spectra of Kf-AQ (Fig. 4a), corresponding to the water clusters including $Na^+(H_2O)_3$ or $OH^-(H_2O)_3$ (3540 cm$^{-1}$), $OH^-(H_2O)_4$ (3410 cm$^{-1}$), and $OH^-(H_2O)_5$ (3292 cm$^{-1}$)[60,61]. These adsorbed water clusters were the proton precursors for $H_2O_2$ photosynthesis, which can be further checked by in-situ Raman spectra. The notable O-H stretching bands in Raman spectra at around 3000–3700 cm$^{-1}$ can be deconvoluted into three bands, corresponding to the four-coordinated hydrogen bonded

water network ($V_1$, ~3254 cm$^{-1}$), the two-coordinated single donor hydrogen bonded water clusters ($V_2$, ~3420 cm$^{-1}$) and the $Na^+$ ion hydrated water ($Na \cdot H_2O$) clusters ($V_3$, ~3553 cm$^{-1}$), respectively (Fig. 4b)[62–64]. The intensity of these bands for Kf-AQ was significantly higher than those for TpAQ, suggesting the formation of stronger hydrogen bond between keto moiety (-C=O) and $OH^-(H_2O)_n$ clusters[65], possibly because the vibrational dipole moment (the direction of O-H bonds) in the clusters (such as $Na \cdot H_2O$) is parallel to the direction of the interfacial electric field, thus favoring the combination of hydrogen in the clusters with carbonyl groups of Kf-AQ. Simultaneously, $V_2$ and $V_3$ were the dominant forms in the Kf-AQ Raman spectrum, and generally had relatively weaker hydrogen bond network than $V_1$, the dominant form in the TpAQ Raman spectrum. These differences can be attributed to the strong interaction between water clusters and carbonyl groups of Kf-AQ, resulting in the disorder and stretching of H-bonds in the arrangement of water molecules[66], and the strong dipole-dipole force between $Na^+$ and $H_2O$ molecules in the $Na^+$ solvation structures further destroy the water-water interactions to form small water clusters of weak H-bonding environment[66], thus favoring the

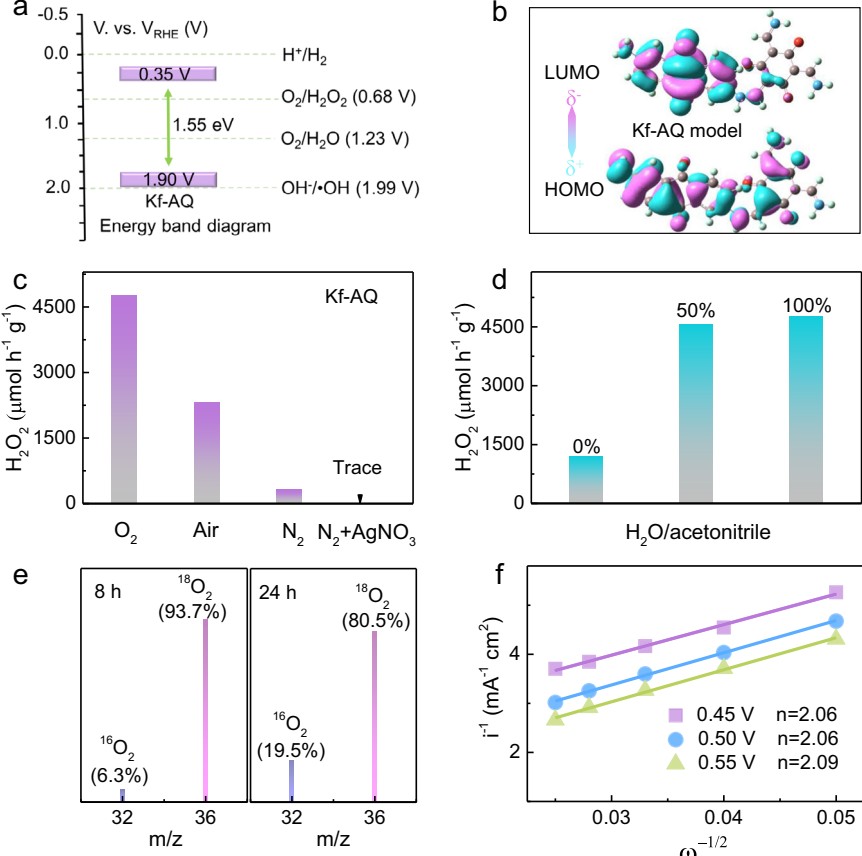

**Fig. 3 | Mechanism investigation. a** Band edge potentials of Kf-AQ. **b** Calculated HOMO and LUMO for Kf-AQ dimer. **c**, **d** A comparison on $H_2O_2$ production rates over Kf-AQ in different atmospheres and solutions. **e** Mass charts for $O_2$ evolved by decomposition of $H_2O_2$ produced at pH 13 by isotopic experiments. **f** The Koutecky-Levich plots of Kf-AQ obtained by RDE measurements.

photocatalytic dissociation of water and release of hydrogen. Therefore, the exclusive keto-form of Kf-AQ enhanced the adsorption and dissociation of water, thereby promoting hydrogen abstraction from water for the $H_2O_2$ photosynthesis.

We compared water adsorption over Kf-AQ and enol-form TpAQ by DFT calculations. In case of one water molecule adsorption, the adsorption energy of Kf-AQ was −0.26 eV, much lower than that of enol-form TpAQ (−0.18 eV) (Fig. S24). Increasing water cluster sizes to $(H_2O)_3$, the adsorption energy of Kf-AQ decreased to −0.35 eV, and further decreased to −0.44 eV for the $OH^-(H_2O)_2$ clusters, which was the dominant form of adsorbed water in alkaline water (Fig. S25), suggesting the strong water adsorption capability of Kf-AQ. Moreover, the bond energy of terminal H-O in $OH^-(H_2O)_2$ form was 4.3 eV, much lower than that of $(H_2O)_3$ (5.9 eV) (Fig. 4c), suggesting the easier hydrogen dissociation from the terminal water, and thus favoring the subsequently combination with the neighboring $H_2O$ to form hydronium ion $(H_3O^+)$[18].

We detected the intermediates of $H^*_{ads}$ and $OH_{ads}$ species by the cyclic voltammogram (CV) (Fig. 4d). The $H^*_{ads}$ species generated in the reduction stage by reducing hydronium ion $(H_3O^+)$ were oxidized, corresponding to an oxidative peak at about 0.25 V vs. RHE[67,68]. The oxidative peaks in the CV curves were more distinct with the increase of hydroxide anion concentrations, suggesting the increase of $H^*_{ads}$ dosages at strong alkaline conditions[18]. Simultaneously, the reduction peak at 0.77 V was attributed to the reversible adsorbed $OH_{ads}$ species, which were produced via the loss of electrons in $OH^-$ (ref. 69). The $OH_{ads}$ species would be stabilized by forming hydroxyl-water-alkali metal cation cluster $(OH_{ads}-Na^+-(H_2O)_n)$, thus accordingly preventing its depletion by $H_3O^+$. Therefore, we propose that the dissociation of

$H_2O$ into $H^*_{ads}$ and $OH_{ads}$ species takes place in the 2e⁻ ORR and 4e⁻ WOR pathways.

These above results strongly suggest a synergism of keto and anthraquinone moieties in Kf-AQ for superior $H_2O_2$ photosynthesis from water and oxygen, as depicted in Fig. 5. Initially, $OH^-(H_2O)_n$ clusters preferentially adsorbs onto the keto-form moieties in Kf-AQ, thus weakening the H-O bond of the terminal $H_2O$ via forming the H-$OH(H_2O)_{n-1}OH^-$ clusters and facilitating the dehydrogenation in water molecules. The detached protons then combine with the neighboring $H_2O$ to form $H_3O^+$, which was proved by the in-situ FTIR spectra of Kf-AQ under alkaline condition (Fig. S26), as the absorption band at 3525 cm⁻¹ for the stretching vibrations of the O-H group in $H_3O^+$ was progressively intensified[69,70]. Upon visible light irradiation, surface $H_3O^+$ on Kf-AQ can be reduced by interfacial electrons (e⁻) to release $H^*_{ads}$ species, which preferentially bind with the quinone groups (-C=O) in AQ and subsequently hydrogenate AQ to yield anthrahydroquinone ($H_2AQ$). Afterwards, the parahydrogen atoms of $H_2AQ$ are abstracted to produce radicals, which react with $O_2$ to form 1,4-endoperoxide species, a well-known intermediate for the formation of $H_2O_2$, which was confirmed by the new Raman peak at 891 cm⁻¹ (Fig. S27). Then, 1,4-endoperoxide species couples the adjacent hydrogen in the hydroxyl group of $H_2AQ$ to release $H_2O_2$. Meanwhile, another dissociation product, $OH_{ads}$ intermediate, would not be dissociated as $OH^-$ within the interface layer, but form an adsorbed hydroxyl-water-alkali metal cation cluster $(OH_{ads}-Na^+-(H_2O)_n)$[69]. Upon visible light irradiation, the photogenerated holes (h⁺) oxidizes this $OH_{ads}$ to produce $O_2$ in a 4e⁻ WOR pathway. Therefore, the formation of $OH_{ads}-Na^+-(H_2O)_n$ and $H_3O^+$ intermediates over Kf-AQ at high pH conditions facilitates water

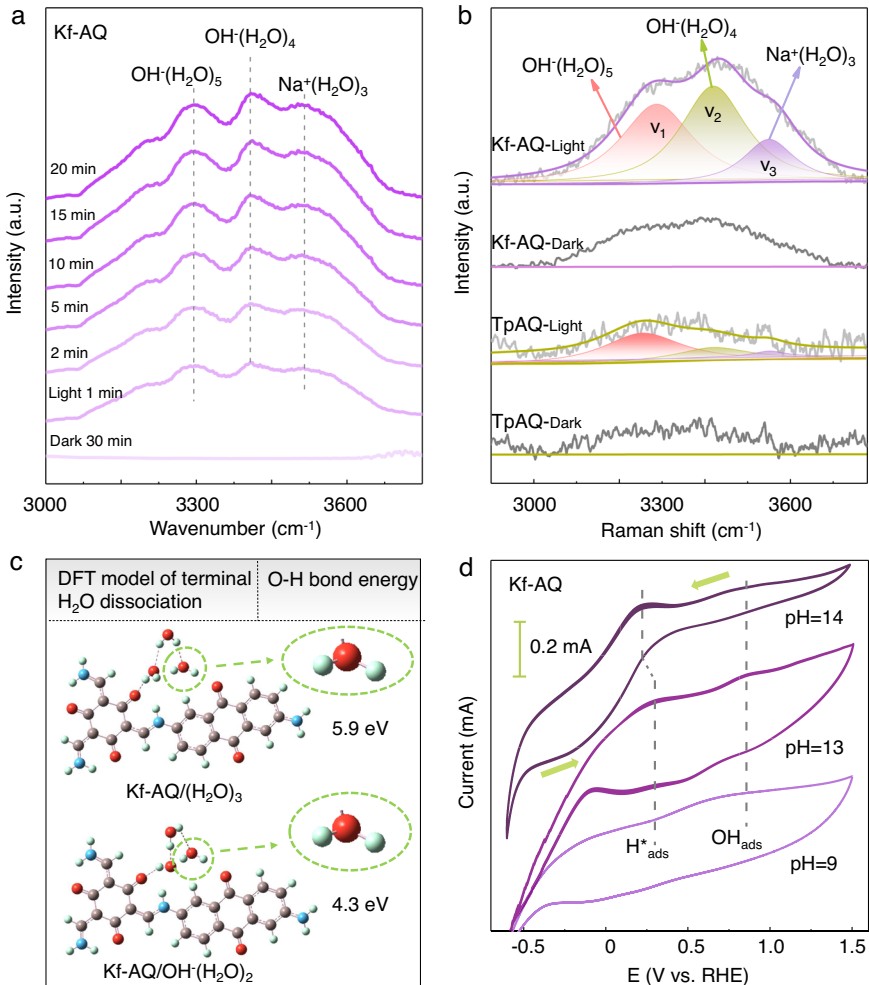

**Fig. 4 | Mechanism investigation. a** In-situ FTIR spectra of Kf-AQ suspension for $H_2O_2$ photosynthesis. **b** Raman spectra of Kf-AQ and TpAQ suspensions under visible light irradiation. **c** The H-O bond energy of the adsorbed terminal $H_2O$ over the Kf-AQ via DFT calculation. **d** Cyclic voltammogram (CV) curves of Kf-AQ in different pH electrolytes.

oxidation and hydrogen extraction from $H_2O$ molecules, resulting in its efficient photocatalytic $H_2O_2$ production.

## Discussion

In summary, we have demonstrated the synthesis of a keto-form anthraquinone containing COF via a mechanochemical process and its efficient $H_2O_2$ photosynthesis in alkaline water, with a record $H_2O_2$ production rate of 4784 µmol h$^{-1}$ g$^{-1}$ under visible light irradiation in the absence of sacrificial reagents. The keto-form structure in Kf-AQ can promote the water adsorption through the formation of $OH^-(H_2O)_n$ clusters with weakened hydrogen bonds, which accordingly enhances the dehydrogenation of water and promotes efficient $H_2O_2$ photosynthesis. The manipulating H-bond network of adsorbed water clusters represents a strategy to break the rate-limiting step of hydrogen extraction from water, and bring insights for the design of highly active photocatalysts to realize efficient $H_2O_2$ photosynthesis from only water and oxygen.

## Methods
### Synthesis of Kf-AQ
Mechanochemical synthesis was conducted to produce Kf-AQ by use of a planetary ball mill (SFM-1, Hefei Kejing Material Technology Co., Ltd). Typically, 2,4,6-triformylphloroglucinol (Tp, 126 mg, 0.20 mmol), 2,6-diaminoanthraquinone (AQ, 213 mg, 0.30 mmol), and $CH_3COONa$ (5 mg) were placed in a 50 mL agate grinding jar, with fifteen 7 mm

diameter and ten 5 mm diameter agate balls. Then, the mixture was ground at room temperature with a rotation speed of 400 rpm for 6 h. After that, the obtained precursors were washed with N, N-dimethylformamide and acetone, and then dried in a vacuum oven at 120 °C for 12 h. The obtained photocatalyst was denoted as Kf-AQ.

### Synthesis of TpAQ
TpAQ was synthesized by Schiff-base condensation of Tp and AQ according to a modified previous method[28]. In a 10 mL Schlenk tube, 2,4,6-triformylphloroglucinol (71.4 mg, 0.20 mmol) and 2,6-diaminoanthraquinone (42.1 mg 0.30 mmol) were charged. Then, N, N-dimethylacetamide (2.0 mL) was added as the solvent, and the suspensions were sonicated for 10 min. Subsequently, 0.3 mL glacial acetic acid was added. After that, the ampoule was degassed by freeze-pump-thaw three times and then sealed on. The Schlenk tube was put into an oven and heated at 120 °C for 72 h. The obtained powder was washed with N, N-dimethylformamide and acetone, and then dried in a vacuum oven at 120 °C for 12 h.

### $H_2O_2$ photosynthesis
$H_2O_2$ photosynthesis was conducted in a homemade quartz cuvette reactor. Generally, 5 mg of the photocatalysts were ultrasonically dispersed into 30 mL water whose initial pH was adjusted by 0.1 M NaOH solution. Then, the suspension was stirred for 15 min in the dark with continuously $O_2$ purging. After that, the reactor was illuminated by a

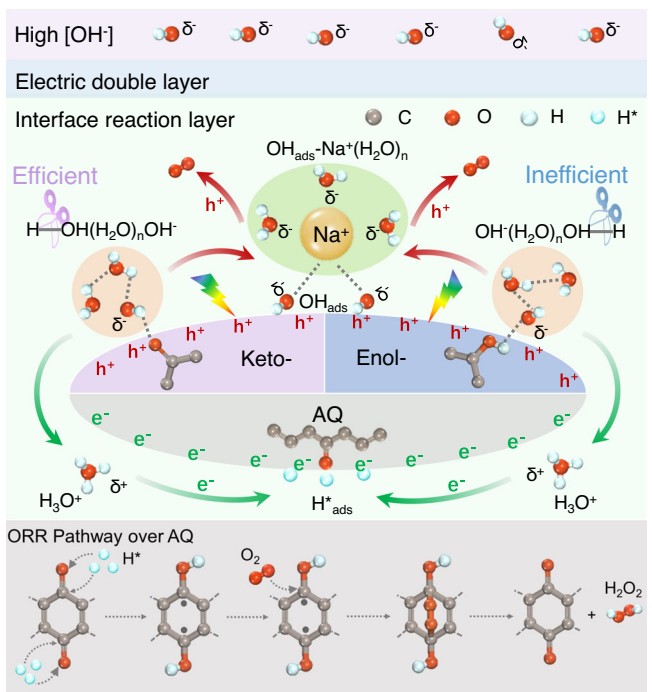

**Fig. 5 | Mechanism of H₂O₂ photosynthesis.** The photocatalytic pathway for H₂O₂ production over Kf-AQ in alkaline conditions.

300 W Xe lamp (PLS-SXE300, Beijing Perfectlight) with a cut-off filter (λ > 400 nm). During the reaction, 1.5 mL of reaction mixture was withdrawn at every 15 min interval, and then filtrated through a 0.45 μm polyether sulfone (PES) filter for $H_2O_2$ detection.

The $H_2O_2$ concentration was measured by the N, N-diethylp-phenylenediamine (DPD)-horseradish peroxidase (POD) colorimetry method[71]. Typically, 3.0 mL of phosphate buffer (0.2 M, pH = 6) was added into a 15.0 mL colorimetric tube, and then 1.0 mL sample, 50 μL DPD, 50 μL POD were added into the mixture. Then, ultrapure water was added to set the volume to 10 mL. Finally, the absorbance was measured on a UV-vis spectrophotometer (TU-1810) at 551 nm to determine $H_2O_2$ concentration according to the predetermined standard curve.

### In-situ FTIR measurements
In-situ FTIR spectra were obtained by using a Thermo Scientific Nicolet Is50, equipped with a commercial chamber from Harrick Scientific. Typically, 5 mg of Kf-AQ was dispersed into 30 mL $H_2O$ at pH = 13. The formed uniform dispersion was bubbled with $O_2$ for 15 min in the dark, and then the background spectrum was collected. After that, the reaction chamber was irradiated by visible light (λ > 400 nm), and then the spectrum was collected at a one min interval.

### Isotopic experiments
Specifically, 5 mg Kf-AQ was added into $H_2^{16}O$ (30 mL) within a glass tube (50 mL). The formed dispersion was sonicated for 10 min and bubbled with Ar for 30 min. Then, the reaction tube was sealed with rubber septum cap and vacuum. $^{18}O_2$ gas was introduced to the tube by a syringe. The reaction tube was illuminated by a 300 W Xe lamp with a cut-off filter (λ > 400 nm). After photoreaction for 8 h and 24 h, the reaction solution was purged by Ar for 5 min to remove the residual $^{18}O_2$ gas. The dispersion was filtered and injected into another clean tube, which was saturated with Ar and contained 200 mg $MnO_2$ powder. The generated gas was collected by an Aluminum foil air pocket (5 mL) and detected on a Shimadzu GC-MS system (Agilent 7890 A/ 5975 C).

### DFT calculations
The DFT calculation used the method in previous ref. 72. Briefly, geometry optimizations without symmetry restriction are performed by using the DFT/B3LYP/6-31 G(d, p) basis sets and scrf-smd solvent model. All calculations were performed on Gaussian 09. The enol-form TpAQ (Fig. S28a) and Kf-AQ (Fig. S28b) dimers were used as calculation models for $H_2O$ molecules and $OH^-(H_2O)_n$ clusters adsorption and dissociation.

The adsorption energy (Eads) of adsorbate (A, indicating $H_2O$ or $OH^-(H_2O)_n$) was defined as Eq. (1), wherein, E (A*), E (*) and E (A) are the energy of A adsorbed on the active site, the energy of active site, and the energy of isolated A, respectively. The O-H bond dissociation energy of $H_2O$ over Kf-AQ dimer was calculated by Eq. (2), wherein, E (Kf-AQ·H₂O) is the energy of $H_2O$ adsorbed on the Kf-AQ, E (OH⁻) is the energy of isolated OH⁻, and E (Kf-AQ·H⁺) is the energy of H⁺ adsorbed on the Kf-AQ.

$$E_{ads} = E(A^*) - E(^*) - E(A) \quad (1)$$

$$E(O-H)_{BDE} = E(Kf - AQ - H_2O) - E(OH^-) - E(Kf - AQ - H^+) \quad (2)$$

Solvation has been a conscientious consideration in our study, and we employed an implicit solvation model for calculations.

### Data availability
The data that support the findings of this study are available from the corresponding author upon request.

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

## Acknowledgements
Financial supports from the National Natural Science Foundation of China (Nos. 52070128, 22206125, 22376138) are gratefully acknowledged.

## Author contributions
X.Z., M.L., B.Z., and L.Z. contributed to design of this study, X.Z., M.L., and L.Z. wrote the manuscript. X.Z., J.M., C.C., and S.C. conducted experiments and performed data analysis. X.W. provided DFT calculation and analysis.

## Competing interests
The authors declare no competing interests.
