## [Peer Review File · Nature Communications]

Keto-anthraquinone covalent organic framework for H₂O₂ photosynthesis with oxygen and alkaline waterREVIEWER COMMENTS

Reviewer #1 (Remarks to the Author):

In this paper, the authors reported a keto-form anthraquinone covalent organic framework (Kf-AQ) for H₂O₂ photosynthesis from oxygen and alkaline water, which is normally limited by the sluggish water oxidation. The authors have done significant work to deal with this limitation by lowering the energy barrier of hydrogen extraction from water clusters through weakening the H-bonded networks in a keto-form COF under alkaline condition. Accompanying with the in-situ characterizations, DFT calculation provided supports for the enhanced charge transfer of keto-form structures, the formation of OH-(H₂O)_n clusters, and the facilitated dissociation of the terminal O-H bonds. This manuscript is well-organized, and the main arguments are well-supported by the experimental and theoretical results. I recommend this paper acceptance after a revision according to the following specific comments.

On the DFT calculation

1. Figure 3b: The authors analyzed the charge distribution on Kf-AQ under excited states, and emphasized the favorable charge transfer behavior of the keto-form structures. I think the authors should also provide such information of the enol-form TpAQ for comparison.
2. Line 272, "We compared water adsorption over Kf-AQ and TpAQ by DFT calculations." In this section, the authors discussed the difference of water adsorption between keto- and enol- forms, and proved the advantages of Kf-AQ for water adsorption and oxidation. As TpAQ is a mixture of keto- and enol- forms, it is better to use "enol-form TpAQ" instead of "TpAQ" in this section. Authors should clarify this in the texts.
3. Methods: More details about DFT calculation should be provided. For example, clarify the calculation formulas for the adsorption energy and the H-O bond energy. Was a hydrogen atom or a proton considered when the calculation of O-H dissociation? Was there a solvation layer established during the calculation of H₂O adsorption?

A minor comment

4. Figure S12a, the main diffraction peak in the XRD patterns of Kf-AQ has not yet been fully identified. They should be re-measured by starting from a smaller angle.

Reviewer #2 (Remarks to the Author):

Hydrogen peroxide photosynthesis is of great significance for solar energy utilization and H₂O₂ decentralization applications. In this manuscript, the authors reported a mechanochemical synthesis of novel Kf-AQ photocatalyst, which can realize efficient H₂O₂ production from oxygen and alkaline water. The enhanced water oxidation performance of Kf-AQ was attributed to its weakened H-bonded networks of water clusters, as revealed by the in-situ Raman and FTIR spectra. This finding can guide the development of high-performance photocatalyst. The mechanism is supported by experimental data and DFT calculation and the manuscript is well organized. I recommend its publication after a minor revision.

1. The mechanochemical synthesis of Kf-AQ is different from the traditional solvothermal synthesis

method of TpAQ. Why did the authors choose NaAc for the Schiff-base condensation? Would NaOH work?

2. Fractional atomic coordinates for the unit cell should be provided for the simulated X-ray diffraction (PXRD) pattern of Kf-AQ.
3. Was the simulated interlayer distance consistent with the space in the HRTEM image in Figure 1d?
4. Figure 3b, DFT calculation was employed to check the charge separation performance of, which should be discussed in combination with experimental data like PL spectra or photocurrent measurements. The comparison between Kf-AQ and TpAQ is also suggested.
5. Fig. 4b, the dark control Raman spectra of Kf-AQ and TpAQ suspensions should be provided for comparison with those under visible light irradiation.

Reviewer #3 (Remarks to the Author):

The manuscript entitled “Keto-anthraquinone covalent organic framework for H₂O₂ photosynthesis with oxygen and alkaline water” provide a keto-form anthraquinone covalent organic framework (Kf-AQ) for photocatalytic H₂O₂ production from O₂ and alkaline H₂O. The authors suggested that the Kf-AQ with keto-form structure is beneficial to photocatalytic H₂O₂ generation. However, the catalysts with the same structure have been reported in Reference: Chemical Engineering Journal 466 (2023) 143085, which also indicated that the keto-form structure is beneficial to photocatalytic H₂O₂ generation. I think the innovation is not enough, and I would not recommend this manuscript be published in Nature Communications, although this work may be the first example of COF for photocatalytic H₂O₂ production under alkaline conditions.

1. As shown in Figure 1b, the powder X-ray diffraction pattern of Kf-AQ is clear, while no diffraction peak appears in Figure S12, whether before or after photoreaction. Also, I don't agree with the author statement that the simulated powder X-ray diffraction (PXRD) pattern of Kf-AQ with eclipsed AA stacking mode agreed well with the experimental data.
2. As the author said that the pore size of Kf-AQ is 2.12 nm, the curve displayed in Figure S5 is typical III isotherm. While, the adsorption/desorption isotherm of the sample with micropore should be I curve in theoretically, which is not consistent with your experimental data.
3. To more clearly compare the difference in the structure between the Kf-AQ and TpAQ COFs, it is recommended to put the ¹³C CP-MAS solid state NMR spectra of Kf-AQ and TpAQ together.
4. The authors mentioned that “the conduction band potential (ECB) of Kf-AQ was accordingly calculated as 0.35 V. Therefore, 2e⁻ ORR (0.68 V vs. RHE) to produce H₂O₂ was thermodynamically feasible for Kf-AQ photocatalysis”. It should be noted that 2e⁻ ORR to produce H₂O₂ by single-step two-electron oxygen reduction (O₂ + 2H⁺ + 2e⁻ → H₂O₂) usually needs to be done under acidic conditions, it should be two-step one-electron oxygen reduction under alkaline conditions.
5. The superiority of keto-form structure is not well represented in the probable reaction mechanism.

Point-by-point Responses to reviewers' comments

Reviewer #1

General comments

In this paper, the authors reported a keto-form anthraquinone covalent organic framework (Kf-AQ) for H₂O₂ photosynthesis from oxygen and alkaline water, which is normally limited by the sluggish water oxidation. The authors have done significant work to deal with this limitation by lowering the energy barrier of hydrogen extraction from water clusters through weakening the H-bonded networks in a keto-form COF under alkaline condition. Accompanying with the in-situ characterizations, DFT calculation provided supports for the enhanced charge transfer of keto-form structures, the formation of OH(H₂O)_n clusters, and the facilitated dissociation of the terminal O-H bonds. This manuscript is well-organized, and the main arguments are well-supported by the experimental and theoretical results. I recommend this paper acceptance after a revision according to the following specific comments.

Response: We thank the reviewer for these positive comments and helpful suggestions.

Specific comments

On the DFT calculation

1. Figure 3b: *The authors analyzed the charge distribution on Kf-AQ under excited states, and emphasized the favorable charge transfer behavior of the keto-form structures. I think the authors should also provide such information of the enol-form TpAQ for comparison.*

Response: Thanks a lot for this helpful comment. We carried out density functional theory (DFT) calculations to further elucidate the charge transfer property of enol-form TpAQ dimer. As depicted in **Figure S16**, the highest occupied molecular orbital (HOMO) of enol-form TpAQ is uniformly distributed on the dimer, while the lowest unoccupied molecular orbital (LUMO) mainly localizes at the AQ moiety and overlaps

with the HOMO, with a portion of LUMO remaining on the benzene ring of the Tp moiety. In the contrast, the HOMO of Kf-AQ dimer uniformly disperses in the whole structure, while the LUMO mainly localized at AQ units, without any residual LUMO on the benzene ring of the Tp moiety (**Fig. 3b**). The results indicate that Kf-AQ is more favorable for the separation and transfer of photogenerated charges.

Figure S16. Calculated HOMO and LUMO for enol-form TpAQ dimer.

Page 10

“We further conducted density functional theory (DFT) calculations to elucidate the exciton dissociation in photocatalysis by using the dimer models of Kf-AQ and enol-form TpAQ. As depicted in Fig. 3b, the highest occupied molecular orbital (HOMO) of Kf-AQ dimer uniformly disperses in the whole structure, while the lowest unoccupied molecular orbital (LUMO) mainly localized at AQ units, without any residual LUMO on the benzene ring of the Tp moiety. Thus, the HOMO-LUMO transition under excitation can redistribute electron density from the Tp moieties to the adjacent AQ units, thus resulting in effective intramolecular charge transfer in Kf-AQ. In the contrast, the HOMO of enol-form TpAQ is uniformly distributed on the dimer, while the LUMO mainly localizes at the AQ moiety and overlaps with the HOMO, with a portion of LUMO remaining on the benzene ring of the Tp moiety (Fig. S16). These results indicate that Kf-AQ is more favorable for the separation and transfer of photogenerated charges.”

2. Line 272, “We compared water adsorption over Kf-AQ and TpAQ by DFT calculations.” In this section, the authors discussed the difference of water

adsorption between keto- and enol- forms, and proved the advantages of Kf-AQ for water adsorption and oxidation. As TpAQ is a mixture of keto- and enol- forms, it is better to use “enol-form TpAQ” instead of “TpAQ” in this section. Authors should clarify this in the texts.

Response: Thanks a lot for this helpful suggestion. We have revised TpAQ into enol-form TpAQ in the main text as follows.

Page 14

“We compared water adsorption over Kf-AQ and enol-form TpAQ by DFT calculations. In case of one water molecule adsorption, the adsorption energy of Kf-AQ was -0.26 eV, much lower than that of enol-form TpAQ (-0.18 eV).”

3. Methods: More details about DFT calculation should be provided. For example, clarify the calculation formulas for the adsorption energy and the H-O bond energy. Was a hydrogen atom or a proton considered when the calculation of O-H dissociation? Was there a solvation layer established during the calculation of H₂O adsorption?

Response: Thank you very much for these helpful comments. We provided calculation details in the section of “DFT calculations” on Page 19 during the revision as follows. We considered a proton for the calculation of O-H dissociation, and there was a solvation layer during the calculation of H₂O adsorption.

“The enol-form TpAQ (Fig. S28a) and Kf-AQ (Fig. S28b) dimers were used as calculation models for H₂O molecules and OH⁻(H₂O)_n clusters adsorption and dissociation.

The adsorption energy (E_{ads}) of adsorbate (A, indicating H₂O or OH⁻(H₂O)_n) was defined as Eq. (1), wherein, $E(A^*)$, $E(*)$ and $E(A)$ are the energy of A adsorbed on the active site, the energy of active site, and the energy of isolated A, respectively. The O-H bond dissociation energy of H₂O over Kf-AQ dimer was calculated by Eq. (2), wherein, $E(Kf-AQ-H_2O)$ is the energy of H₂O adsorbed on the Kf-AQ, $E(OH^-)$ is the energy of isolated OH⁻, and $E(Kf-AQ-H^+)$ is the energy of H⁺ adsorbed on the Kf-AQ.

$$E_{ads} = E(A^*) - E(*) - E(A) \quad (1)$$

$$E(O-H)_{BDE} = E(Kf-AQ-H_2O) - E(OH^-) - E(Kf-AQ-H^+) \quad (2)$$

Solvation has been a conscientious consideration in our study, and we employed an implicit solvation model for calculations.”

Figure S28. DFT calculation models of enol-form TpAQ and Kf-AQ.

A minor comment

4. Figure S12a, the main diffraction peak in the XRD patterns of Kf-AQ has not yet been fully identified. They should be re-measured by starting from a smaller angle.

Response: We are sorry for this mistake. During the revision, we updated the PXRD patterns in **Figure S12**, which was consistent with that in the **Fig. 1b**.

Figure S12a. The PXRD patterns of Kf-AQ before and after H₂O₂ photosynthesis.

Reviewer #2

General comments

Hydrogen peroxide photosynthesis is of great significance for solar energy utilization and H₂O₂ decentralization applications. In this manuscript, the authors reported a mechanochemical synthesis of novel Kf-AQ photocatalyst, which can realize efficient H₂O₂ production from oxygen and alkaline water. The enhanced water oxidation performance of Kf-AQ was attributed to its weakened H-bonded networks of water clusters, as revealed by the in-situ Raman and FTIR spectra. This finding can guide the development of high-performance photocatalyst. The mechanism is supported by experimental data and DFT calculation and the manuscript is well organized. I recommend its publication after a minor revision.

Response: We highly appreciate the reviewer's positive comments and helpful suggestions.

Specific comments

1. The mechanochemical synthesis of Kf-AQ is different from the traditional solvothermal synthesis method of TpAQ. Why did the authors choose NaAc for the Schiff-base condensation? Would NaOH work?

Response: Thanks a lot for this comment. We found that the solvothermal method could synthesize TpAQ mixture of both keto- and enol-forms after many tentative tests.

As HAc is a typical catalyst for Schiff-base condensation and used in solvothermal method for TpAQ synthesis, and Ac⁻ anions are well-known Lewis base, we thus selected NaAc as the catalyst for the mechanochemical synthesis of Kf-AQ. In the mechanochemical synthesis, Ac⁻ anions tend to bind with the hydrogen of hydroxyl group in enol moieties, and then induce electron transfer from oxygen to alkene group, thus enabling the enol-form transformation into the thermodynamically more stable keto-form structure (**Fig. S10**). Such transformation cannot be realized in the solvothermal synthesis due to the strong intramolecular hydrogen bonding in HAc, making it difficult to ionization of Ac⁻ and then to weaken the O-H bond in the

neighboring enol-form.

We also found that NaOH did work as a catalyst for Tp and AQ condensation in mechanochemical synthesis, but resulting in insufficient conversion from enol-form to keto-form. As expected, the catalyst synthesized by using NaOH as the catalyst exhibited much worse H₂O₂ photosynthesis performance (**Figure R1**).

Figure R1. H₂O₂ photosynthesis over Kf-AQ and Tp+AQ+NaOH.

2. Fractional atomic coordinates for the unit cell should be provided for the simulated X-ray diffraction (PXRD) pattern of Kf-AQ.

Response: Thank you very much for this constructive suggestion. The geometry of Kf-AQ with P6/m symmetry was optimized by MS DMol3. The lattice model was geometrically optimized by using MS Forcite molecular dynamics module, and Pawley refinement was applied to define the lattice parameters and produce the simulated PXRD profile. The fractional atomic coordinate data for the unit cell of Kf-AQ with eclipsed AA stacking were presented in **Table S1**.

Table S1. Fractional atomic coordinates for the unit cell of Kf-AQ.

Kf-AQ (eclipsed AA stacking)			
Space group: P6/m			
a = 30.59 Å, b = 30.59 Å, c = 3.51 Å			
α = 90.00, β = 90.00, γ = 120.00			
C	0.37164	0.67137	0.00259
C	0.35803	0.61843	0.00258
C	0.30756	0.58083	0.00258

C	0.26807	0.59421	0.00259
C	0.29526	0.53077	0.00257
C	0.33242	0.51808	0.00256
C	0.38262	0.55534	0.00255
C	0.39514	0.60527	0.00256
N	0.21828	0.72228	0.00263
N	0.42143	0.54331	0.00254
O	0.41533	0.70392	0.00259
O	0.22438	0.56176	0.00259

Page 5

“The simulated powder X-ray diffraction (PXRD) pattern of Kf-AQ with eclipsed AA stacking mode, whose fractional atomic coordinate data for the unit cell were presented in Table S1, agreed with the experimental data in a large extent (Fig. 1b), suggesting the validity of such structure in Kf-AQ.”

3. Was the simulated interlayer distance consistent with the space in the HRTEM image in Figure 1d?

Response: Thanks a lot for this kind comment. We analyzed the HRTEM image in **Fig.1d** by using Digitalmicrograph software, and found that the lattice spacing of Kf-AQ was 0.33 nm, very close to the simulated interlayer spacing (0.348 nm). This consistence between the HRTEM measurement and the theoretically simulated interlayer spacing suggests the validity of the proposed model. We accordingly updated the texts during the revision.

Fig. 1d HRTEM image of Kf-AQ.

4. Figure 3b, DFT calculation was employed to check the charge separation performance of, which should be discussed in combination with experimental data like PL spectra or photocurrent measurements. The comparison between Kf-AQ and TpAQ is also suggested.

Response: We highly appreciate this constructive comment. We collected the fluorescence spectroscopy of TpAQ and Kf-AQ to compare their charge separation performance. For steady-state fluorescence spectra, Kf-AQ displayed much weaker fluorescence intensity than TpAQ (**Figure S17a**). The time-resolved fluorescence analysis also showed that Kf-AQ had a longer relaxation time of electrons (6.82 ns) than TpAQ (6.07 ns) (**Figure S17b**). The PL spectra results validated the better performance of Kf-AQ in photogenerated-charge separation.

Figure S17. The steady-state (a) and transient-state (b) fluorescence spectra of TpAQ and Kf-AQ.

Page 10

“The fluorescence spectra of TpAQ and Kf-AQ provided further evidence for their charge separation performance. Kf-AQ displayed much weaker fluorescence intensity than TpAQ in the steady-state fluorescence spectra (Fig. S17a), and the time-resolved fluorescence analysis showed that Kf-AQ had a longer relaxation time of electrons (6.82 ns) than TpAQ (6.07 ns) (Fig. S17b), thus proving the better performance of Kf-AQ in photogenerated-charge separation.”

5. Fig. 4b, the dark control Raman spectra of Kf-AQ and TpAQ suspensions should be provided for comparison with those under visible light irradiation.

Response: Thanks for this helpful suggestion. We provided the dark control Raman spectra of Kf-AQ and TpAQ in **Fig. 4b** during the revision.

Fig. 4b Raman spectra of Kf-AQ and TpAQ suspensions under dark and visible light irradiation.

Reviewer #3

General comments

*The manuscript entitled "Keto-anthraquinone covalent organic framework for H_2O_2 photosynthesis with oxygen and alkaline water" provide a keto-form anthraquinone covalent organic framework (Kf-AQ) for photocatalytic H_2O_2 production from O_2 and alkaline H_2O . The authors suggested that the Kf-AQ with keto-form structure is beneficial to photocatalytic H_2O_2 generation. However, the catalysts with the same structure have been reported in Reference: *Chemical Engineering Journal* 466 (2023) 143085, which also indicated that the keto-form structure is beneficial to photocatalytic H_2O_2 generation. I think the innovation is not enough, and I would*

not recommend this manuscript be published in Nature Communications, although this work may be the first example of COF for photocatalytic H₂O₂ production under alkaline conditions.

Response: We highly appreciate the reviewer for pointing out that this is the first example of COF for photocatalytic H₂O₂ production under alkaline conditions.

We would like to further emphasize the innovation of this work in three aspects.

First, we used TpAQ for photocatalytic H₂O₂ production in pure water (*Chemical Engineering Journal* 466 (2023) 143085, ref. 24), and just noticed the critical role of anthraquinone moieties on the H₂O₂ photosynthesis. We did observe the tautomerism of keto-form and enol-form structures in TpAQ-COF, because extending the solvothermal reaction time led to the conversion of the keto-form structure into the enol-form structure. However, the relationship between the COF structures and their performance of H₂O₂ photosynthesis is still unknown, especially the indispensable contribution of keto-form COF to H₂O₂ photosynthesis, which is the main innovation of this study.

Second, Kf-AQ is totally different from TpAQ in both the structure and the synthetic method. TpAQ is a mixed tautomerism of keto-form and enol-form structures that is traditionally synthesized by the solvothermal method. Impressively, Kf-AQ is an exclusive keto-form COF that can only be obtained by the mechanochemical method developed in this study. This is also the first mechanochemical synthesis of Kf-AQ.

Third, as mentioned by the reviewer, this is the first report that the COF with keto-form exhibits superior performance in photocatalytic H₂O₂ production from oxygen and alkaline water. We elucidated the fundamental role of keto-form structures on the facilitated H₂O₂ photosynthesis by promoting hydrogen dissociation in water clusters.

Specific comments

1. As shown in Figure 1b, the powder X-ray diffraction pattern of Kf-AQ is clear, while no diffraction peak appears in Figure S12, whether before or after photoreaction. Also, I don't agree with the author statement that the simulated powder X-ray

diffraction (PXRD) pattern of Kf-AQ with eclipsed AA stacking mode agreed well with the experimental data.

Response: We are sorry for this mistake. During the revision, we updated the PXRD patterns in **Figure S12**, which was consistent with that in the **Fig. 1b**.

Figure S12a. The PXRD patterns of Kf-AQ before and after H₂O₂ photosynthesis.

Regarding the consistence between the experimental and the simulated PXRD pattern of Kf-AQ with eclipsed AA stacking mode, we re-tested and analyzed the PXRD pattern of Kf-AQ, and reworded this argument. The diffraction peak at around 3.11° in the measured pattern, which in our previous version was somewhat compromised due to data processing amplification, is consistent with the strongest peaks in the simulated pattern. Other peaks at 10.3°, 12.1° and 26.54° in the experimental pattern are also found in the simulated pattern with slight shifts, suggesting that the simulated powder X-ray diffraction (PXRD) pattern of Kf-AQ with eclipsed AA stacking mode agreed in a large extent with the experimental data. Such agreements of the experimental and simulated patterns were also reported previously (*J. Am. Chem. Soc.* 2019, 141, 9623-9628). We accordingly updated **Fig. 1b** and the description of “The simulated powder X-ray diffraction (PXRD) pattern of Kf-AQ with eclipsed AA stacking mode agreed with the experimental data in a large extent (Fig. 1b).” on Page 5 in the revised manuscript.

Fig. 1b PXR D patterns of Kf-AQ, experimentally observed (dark) and simulated using eclipsed AA-stacking (red); the inset is the crystal structures of the eclipsed AA stacking model, the simulated cell parameters of a, b and c are 30.59, 30.59 and 3.51 Å, respectively.

2. As the author said that the pore size of Kf-AQ is 2.12 nm, the curve displayed in Figure S5 is typical III isotherm. While, the adsorption/desorption isotherm of the sample with micropore should be I curve in theoretically, which is not consistent with your experimental data.

Response: Thanks a lot for this helpful comment. We are sorry for this mistake in **Figure S5**. During the revision, we re-tested and analyzed the isotherm type and the pore size distribution of Kf-AQ. As depicted in **Figure S5**, the N₂ adsorption-desorption isotherm indicated a mesoporous type IV for Kf-AQ, which was consistent with previous reports (*J. Am. Chem. Soc.* 2013, **135**, 16821-16824). The Brunauer-Emmett-Teller (BET) specific surface area was 141.7 m² g⁻¹, and the prominent pore size distribution peak was about 2.2 nm. The pore size distribution was close to the simulated values (2.28 nm) for eclipsed AA stacking model of Kf-AQ. The information was updated on Page 5 in the revised manuscript.

Figure S5. (a) N₂ adsorption-desorption isotherm and (b) pore size distribution of Kf-AQ.

3. To more clearly compare the difference in the structure between the Kf-AQ and TpAQ COFs, it is recommended to put the ¹³C CP-MAS solid state NMR spectra of Kf-AQ and TpAQ together.

Response: Thank you very much for this helpful suggestion. We merged the NMR spectra of Kf-AQ and TpAQ, and revised **Fig. 1e**, **Fig. S1a**, and the corresponding texts. As shown in the NMR spectra, TpAQ exhibited a distinct enol carbon structure with the peaks at 179 ppm (a') and 101 ppm (f'), while these signals were negligible in Kf-AQ, demonstrating the exclusive keto-form in the Kf-AQ.

Fig. 1e ^{13}C CP-MAS solid state NMR spectra of Kf-AQ and TpAQ.

Page 5

“The chemical shifts at 184 ppm, 144 ppm and 109 ppm in ^{13}C NMR spectra were all indexed to the keto-form structure of Kf-AQ, while the chemical shift at 179 ppm (a') and 101 ppm (f') for the enol-form carbon and at 4.4 ppm for the enol-form hydrogen (c, C-OH) was absent in the ^{13}C and ^1H NMR spectra, respectively (Fig. 1e and Fig. S6).”

4. The authors mentioned that “the conduction band potential (ECB) of Kf-AQ was accordingly calculated as 0.35 V. Therefore, $2e^-$ ORR (0.68 V vs. RHE) to produce H_2O_2 was thermodynamically feasible for Kf-AQ photocatalysis”. It should be noted that $2e^-$ ORR to produce H_2O_2 by single-step two-electron oxygen reduction ($\text{O}_2 + 2\text{H}^+ + 2e^- \rightarrow \text{H}_2\text{O}_2$) usually needs to be done under acidic conditions, it should be two-step one-electron oxygen reduction under alkaline conditions.

Response: Thanks a lot for this constructive comment. We agree with the reviewer that the $2e^-$ ORR to produce H_2O_2 ($\text{O}_2 + 2\text{H}^+ + 2e^- \rightarrow \text{H}_2\text{O}_2$) needs protons and usually is done under acidic conditions. Although our reaction conducted in alkaline solution, protons can be locally produced on the surface to enable $2e^-$ ORR for the formation of H_2O_2 , which were recently reported by several groups (please see *Nat. Commun.* 2019,

10, 4876; *Nat. Commun.* 2023, **14**, 4209).

During the H_2O_2 photosynthesis over Kf-AQ, the weakened hydrogen bonds in the $\text{OH}\cdot(\text{H}_2\text{O})_n$ clusters induces the dissociation of end H_2O molecules to form H_3O^+ , which was confirmed by the in-situ FTIR spectra. As depicted in **Figure S26**, the FTIR spectra of Kf-AQ under alkaline condition display a progressively intensified absorption band at 3525 cm^{-1} , which is ascribed to the stretching vibrations of the O-H group in H_3O^+ (please see *Nat. Commun.* 2022, **13**, 2024; *Adv. Energy Mater.* 2023, **13**, 2203136). The formation of H_3O^+ could create a unique localized acidic microenvironment (please see *Nat. Commun.* 2019, **10**, 4876; *Nat. Commun.* 2023, **14**, 4209), leading to a possible single-step two-electron reaction to produce H_2O_2 .

Figure S26. In-situ FTIR spectra of Kf-AQ under alkaline condition.

Page 16

“The detached protons then combine with the neighboring H_2O to form H_3O^+ , which was proved by the in-situ FTIR spectra of Kf-AQ under alkaline condition (Fig. S26), as the absorption band at 3525 cm^{-1} for the stretching vibrations of the O-H group in H_3O^+ was progressively intensified^{48, 49}.”

References:

48 Tan, H. et al. Engineering a local acid-like environment in alkaline medium for efficient hydrogen evolution reaction. *Nat. Commun.* **13**, 2024 (2022).

49 Yu, W. et al. High-density frustrated lewis pair for high-performance hydrogen evolution. *Adv. Energy Mater.* **13**, 2203136 (2023).

5. The superiority of keto-form structure is not well represented in the probable reaction mechanism.

Response: Thanks for this comment. We modified the reaction mechanism diagram to highlight the important of keto-form structure in the H_2O_2 photosynthesis during the revision as follows.

Fig. 5 The photocatalytic pathway for H_2O_2 production over Kf-AQ in alkaline conditions.

REVIEWERS' COMMENTS

Reviewer #1 (Remarks to the Author):

The questions from this reviewer have addressed properly after the revision. I recommend the acceptance of the present manuscript.

Reviewer #2 (Remarks to the Author):

The author has responded to all previous questions. I suggest this manuscript to be accepted for publication in Nature Communications.

Reviewer #3 (Remarks to the Author):

I have read the revised manuscript, and the revised manuscript has been modified in accordance with the requirements. I suggest to accept it for publication.